# The Thixoforming Process Window for Al-Si-Zn Alloys Using the Differentiation Method: The Role of Si, Heating Rate and Sample Mass

**Daimer Velazquez Tamayo** [1,2], **Gabriela Lujan Brollo** [2], **Juliana Rodrigues de Oliveira** [2], **Fabio Miani** [3] **and Eugênio José Zoqui** [2,*]

1   Faculty of Mechanical Engineering, University de Oriente (UO), 25VM+C34,
    Santiago de Cuba 90500, Cuba; dvelazqueztamayo@gmail.com
2   Materials and Manufacturing Department, Faculty of Mechanical Engineering, University of Campinas,
    UNICAMP, Campinas 13083-860, SP, Brazil; gbrollo@fem.unicamp.br (G.L.B.);
    jroliveira@fem.unicamp.br (J.R.d.O.)
3   Polytechnic Department of Engineering and Architecture, University of Udine, Via delle Scienze 208,
    33100 Udine, Italy; fabio.miani@uniud.it
*   Correspondence: zoqui@fem.unicamp.br

**Abstract:** The effects of Si content (X = 4 to 7 wt.%), heating rate (5 to 25 °C/min) and sample mass (20 to 200 mg) on determination of the thixoforming working window by differential scanning calorimetry DSC were analyzed for the Al-Xwt.%Si-4wt.%Zn, or simply $Al_XSi_4Zn$, system. The critical lower and upper temperatures for thixoforming processing were determined by applying the differentiation method to DSC heating cycle data. Lower Si content, heating rate and DSC sample mass made identification of the working window temperatures more accurate because of the sharpening of the DSC curve when lower values of these variables were used. Data obtained when lower sample masses and heating rates were used agreed better with those obtained by Calculation of Phase Diagrams, (CALPHAD) simulation (near-equilibrium Scheil condition) for all the Si contents analyzed. Larger DSC sample masses were associated with significant heterogeneity in heat transfer through the sample, leading to results similar to those for a diffuse transition, an effect enhanced by an increase in the heating rate. Since Si content represented a limitation when identifying the working window by the differentiation method, alloys with high Si content should be analyzed with lower DSC masses and lower heating rates to allow more accurate determination of the interval at conditions near those used in thixoforming operations.

**Keywords:** CALPHAD; thixoforming working window; DSC; Al-Si-Zn alloys

## 1. Introduction

Determining the extent to which the solid-to-liquid transformation can be controlled, i.e., the thixoformability of a material, is the first step in evaluating potential feedstock for use in semisolid materials processing (SSM). In the processing of materials in the semisolid state, the main issue to take into account is the thermodynamic control capacity of the operations.

In recent years, different criteria have been developed to assess whether the alloy of interest is applicable for semisolid processing, in other words, whether there is an acceptable process window. Important contributions of thixoformability criteria include the works of Liu et al. [1], Kazakov [2], Tzimas and Zavaliangos [3], Zoqui et al. [4] and recently Hu X.G. et al. [5]. These criteria are divided into three types: the temperature sensitivity of the liquid (solid) fraction [2], the time sensitivity of the liquid fraction [6] and the enthalpy sensitivity of the liquid fraction [5]. However, when it comes to specific alloys, neither enthalpy sensitivity nor time sensitivity provide a clear basis for judgment [7]. In

this sense, the temperature sensitivity of the liquid (solid) fraction ($df_L/dT$ or $df_S/dT$ in $°C^{-1}$), to date, is the most used [7]. Therefore, the criteria adopted in this paper is as follows: "The sensitivity of the liquid fraction (or solid fraction—$df_L/dT$ or $df_L/dT$), at the desired liquid fraction, (or solid fraction—$f_L$ or $f_S$), exclusively for the primary phase must be as low as possible (<0.03 $°C^{-1}$)" [4].

Thixoformability is determined by a set of thermodynamic characteristics, which are, in turn, mainly determined by the chemical composition of the alloy, but are also affected by the processing conditions to which the alloy is subjected, e.g., controlled heat input for endothermic processing, which involves all types of thixoforming operations [4], or even controlled heat extraction for exothermic processing (rheocasting processes) [8], as well as by kinetic effects (e.g., heating rate) [9].

The main tool used to assess thixoformability is the solid fraction vs. the temperature ($f_S$ vs. $T$) curve of the alloy in the semisolid state. This can be determined by CALPHAD (Calculation of Phase Diagrams) simulation [4,10–14] (at equilibrium/near-equilibrium conditions for solidification) or experimentally (far from equilibrium conditions and during melting) by techniques such as differential scanning calorimetry (DSC) using low-mass samples (of the order of milligrams) to provide rapid, reliable measurement of the heat flow variations associated with the semisolid transformations [4,10–14]. In this paper, the effects of Si content (X = 4–7 wt.%), heating rate (5–25 °C/min) and sample mass (20–200 mg) on determination of the thixoforming working window by DSC are analyzed for the Al-Xwt.%Si-4wt.%Zn ($Al_XSi_4Zn$) system using the differentiation method (DM) [15,16] in an attempt to increase the range of raw materials that can be used in rheocasting and thixoforming operations, as well as to provide a better understanding of the practical choices that must be made when selecting the heating rate and sample size in DSC experiments. Note that the processing window evaluation performed in this work refers to thixoforming processes and is not suitable for rheocasting processes.

## 2. Materials and Methods

Thermodynamic simulation and thermal analysis were performed with four $Al_XSi_4Zn$ alloys produced by conventional casting from a mix of raw materials including A356 alloy and commercially pure Al and Zn. These were melted in the appropriate proportions in a SiC crucible in an electric furnace, built in the laboratory, at 700 °C and then poured into metallic molds. An acceptable deviation of ±0.3 wt.%Si and ±0.2 wt.%Zn was adopted. The chemical composition was determined with a BILL OES optical spectrophotometer (Anacon Scientific, São Paulo, Brazil) and is shown in Table 1.

**Table 1.** $Al_XSi_4Zn$ alloys (composition in wt.%) for use in thixoforming processes; standard uncertainties are shown as the deviation corresponding to a 0.95 confidence interval.

| Alloy | Si | Zn | Fe | Mg [1] | Cu [1] | Mn [1] | Ti [1] | Res. [1,2] | Al |
|---|---|---|---|---|---|---|---|---|---|
| $Al_4Si_4Zn$ | 4.15 ± 0.20 | 3.88 ± 0.13 | 0.17 ± 0.02 | 0.14 | 0.08 | 0.07 | 0.03 | 0.03 | Bal. [3] |
| $Al_5Si_4Zn$ | 5.18 ± 0.16 | 3.98 ± 0.18 | 0.19 ± 0.02 | 0.17 | 0.10 | 0.09 | 0.04 | 0.04 | Bal. [3] |
| $Al_6Si_4Zn$ | 6.11 ± 0.28 | 3.86 ± 0.16 | 0.19 ± 0.03 | 0.23 | 0.12 | 0.11 | 0.06 | 0.06 | Bal. [3] |
| $Al_7Si_4Zn$ | 7.24 ± 0.17 | 3.72 ± 0.14 | 0.19 ± 0.02 | 0.39 | 0.14 | 0.12 | 0.07 | 0.05 | Bal. [3] |

[1] When not specified, standard deviation is ≤0.01. [2] Sum of residual elements, such as Cr, Ni and Pb. [3] Balance.

Thermodynamic simulation was performed with Thermo-Calc® software (V4, Basel, Switzerland) and the TTL5 database, which was used to generate the solid fraction ($f_S$) vs. temperature ($T$) curve (where $f_S$ is dimensionless and $T$ is in °C) for all the alloys under the non-equilibrium Scheil condition [15,16]. The percentages of Al, Si, Zn, Mg, Fe, Cu, Mn, Ti, Ni and Cr were taken into account in the simulation.

Thermal analysis to determine the experimental heat flow rate ($HFR$) vs. $T$ curves for the alloys ($HFR$ in mW/mg) was performed with a NETZSCH STA 409 C system, NETZSCH-Gerätebau GmbH Büro Leuna, Germany (with a measurement accuracy of more than 0.5 °C) using solid cylindrical samples weighing approximately 20, 90 and 200 mg.

These were chosen to cover a wide range of masses compatible with the dimensional limitations of the DSC apparatus. The DSC samples were heated to 700 °C at 5, 10, 15, 20 and 25 °C/min. The faster heating rates analyzed here are close to those normally used in industrial thixoforming processes [15,16]. The cooling curves, more suitable for the evaluation of the rheocasting processing window, will be analyzed in a future work.

For each alloy and sample mass, all the thermoanalytical tests were carried out with the same specimen to avoid differences in chemical composition along the billet influencing the results. Prior to data acquisition, the DSC samples were heated to 700 °C and then cooled to room temperature to ensure that the composition was uniform throughout and to avoid incipient melting/solidification in subsequent cycles [15,16].

Thixoformability analysis was performed using the differentiation method (DM) [15] by calculating the derivative of the *HFR* data with respect to temperature to give the variation in the *HFR* with temperature during heating (i.e., the *dHFR/dT* vs. *T* curves, where *dHFR/dT* is in mW/mg°C and *T* is in °C). The DM allowed several critical temperatures, such as the solidus and liquidus temperatures, as well as the lower ($T_{SSML}$) and upper ($T_{SSMH}$) limits of the thixoforming working window ($\Delta T_{THIXO}$), to be identified. The same method can be applied to determine the critical temperatures for rheocasting processes [15,16]. The $f_S$ vs. *T* relationship, where $f_S$ is dimensionless and *T* is in °C, was then determined with NETZSCH Protheus® thermal analysis software (V5, NETZSCH-Gerätebau GmbH Büro Leuna, Germany) by applying the Flynn method [17] of integration of partial areas between the DSC curves and baseline using the values of the solidus and liquidus as references.

The sensitivity curve (i.e., $df_S/dT$ vs. *T*, where $df_S/dT$ is in °C$^{-1}$ and *T* in °C) was also plotted to analyze the stability of $\Delta T_{THIXO}$. Origin® V4 software (OriginLab Corporation, Northampton, MA United States) was used for further integration and differentiation operations and to plot the graphs [4,13,15,16].

The angle ($\theta$) between the baseline and DSC curve at the beginning of the melting transformation, i.e., on the left of the curves, was measured for all the cycles studied. According to Braga et al. [18], $\theta$ is proportional to the total thermal resistance ($R$) associated with the DSC assay, and the relationship can be expressed as $\tan(\theta) = 1/R$. The total resistance, in turn, is the sum of three resistances, *RI*, *RC* and *RS*, which are associated, respectively, with the DSC instrument, the crucible and the sample. The instrument and crucible are the same in all the cycles studied, so any change in R observed here is expected to be due to the change in the sample mass (*RS*).

The integral of the *HFR* (in mW/mg) with respect to temperature (in °C) gives the area (in mW°C/mg) between the DSC curves and a baseline (represented here as *HFR* vs. *T*). The integral of the *HFR* (in mW/mg) with respect to time (s) gives the enthalpy change, $\Delta H$ (in J/g), corresponding to the melting transformation.

## 3. Results

The DSC curves for all the alloys, heating rates and sample masses studied here are shown in Figure 1. The curves have two endothermic valleys, one related to the melting of the Si-rich eutectic, at lower temperatures (left), and another related to the melting of the Al-rich primary phase (Al-$\alpha$), at higher temperatures (right). Each valley is indicated in the curves in Figure 1a for reference. The area between the DSC curves and the baseline, the enthalpy change corresponding to the melting reaction and the solidus, liquidus and semisolid interval for all the cycles are shown in Tables 2–4, respectively. Table 5 compares the variation in these parameters for extreme (lower and upper) values of Si content, heating rate and sample mass.

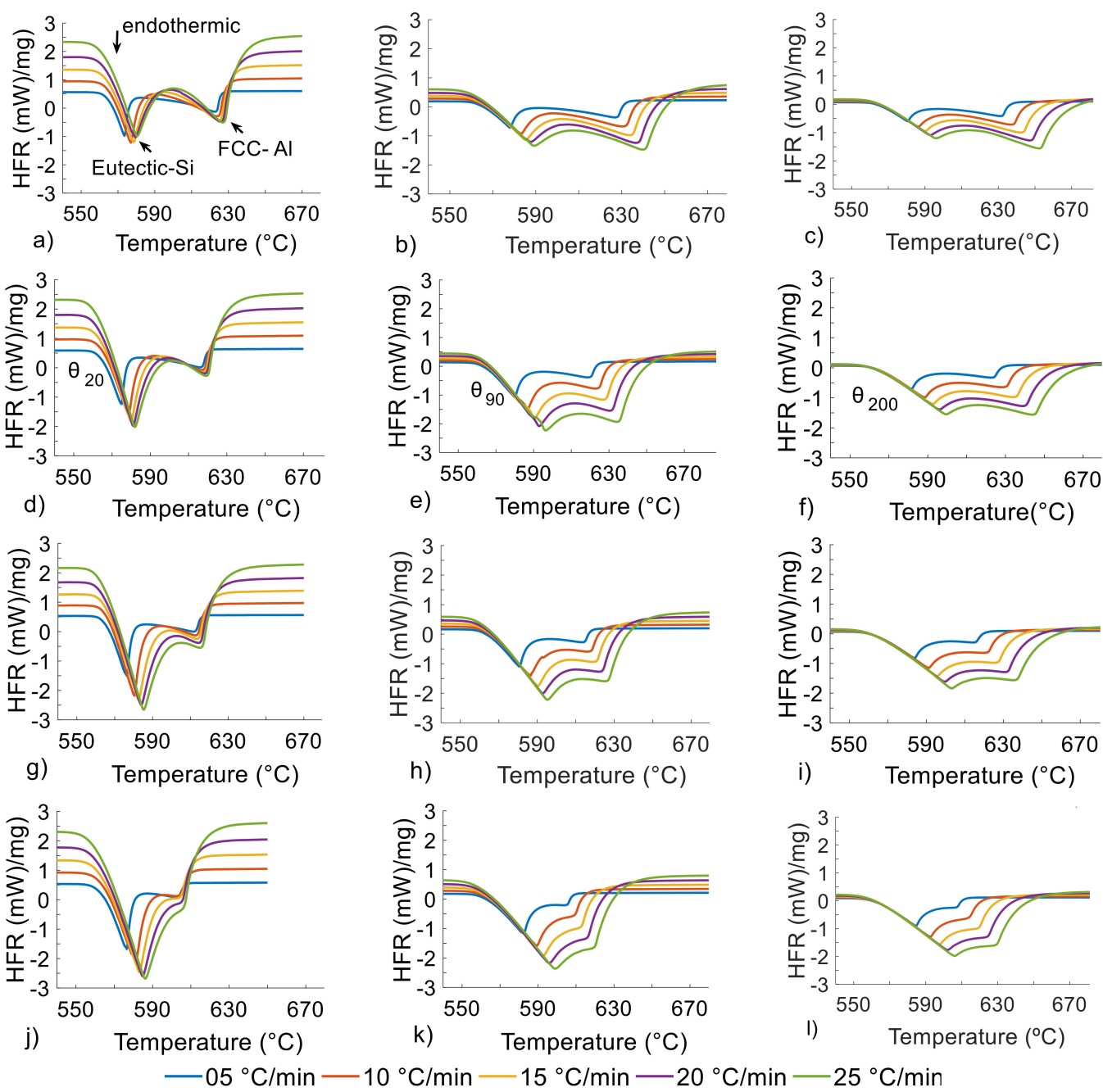

**Figure 1.** DSC heat flow rate (*y*-axis) vs. temperature (*x*-axis) curves during melting of the $Al_XSi_4Zn$ alloys, X = 4 (**a–c**), 5 (**d–f**), 6 (**g–i**) and 7 wt.%Si (**j–l**), at 5–25 °C/min with DSC sample masses of 20 (**left**), 90 (**middle**) and 200 mg (**right**).

**Table 2.** Area between the DSC curves and baseline for the alloys studied here when heated at 5 and 25 °C/min with different sample masses.

|  | Area between DSC Curve and Baseline (W°C/g) | | | | | |
|---|---|---|---|---|---|---|
| *dT/dt* (°C/min) | 5 | - | - | 25 | - | - |
| Mass (mg) | 20 | 90 | 200 | 20 | 90 | 200 |
| $Al_4Si_4Zn$ | 30.6 | 27.9 | 25.5 | 162.5 | 140.1 | 121.3 |
| $Al_5Si_4Zn$ | 30.8 | 31.1 | 24.5 | 167.4 | 159.0 | 122.1 |
| $Al_6Si_4Zn$ | 31.8 | 29.0 | 25.4 | 172.8 | 144.9 | 122.8 |
| $Al_7Si_4Zn$ | 32.6 | 28.9 | 24.9 | 163.9 | 139.4 | 119.1 |

**Table 3.** Enthalpy change for the melting transformation for the alloys studied here when heated at 5 and 25 °C/min with different sample masses.

| | Enthalpy Change (J/g) | | | | | |
|---|---|---|---|---|---|---|
| *dT/dt* (°C/min) | 5 | - | - | 25 | - | - |
| Mass (mg) | 20 | 90 | 200 | 20 | 90 | 200 |
| Al$_4$Si$_4$Zn | 369.7 | 334.8 | 309.5 | 364.8 | 312.9 | 275.8 |
| Al$_5$Si$_4$Zn | 381.6 | 375.5 | 295.7 | 361.0 | 360.9 | 277.8 |
| Al$_6$Si$_4$Zn | 388.0 | 349.0 | 303.9 | 389.8 | 324.3 | 279.0 |
| Al$_7$Si$_4$Zn | 394.0 | 344.0 | 294.0 | 359.0 | 313.8 | 272.8 |

**Table 4.** Solidus, liquidus and semisolid interval for the melting transformation for the alloys studied here when heated at 5 and 25 °C/min with different sample masses.

| | Solidus Temperature (°C) | | | | | |
|---|---|---|---|---|---|---|
| *dT/dt* (°C/min) | 5 | - | - | 25 | - | - |
| Mass (mg) | 20 | 90 | 200 | 20 | 90 | 200 |
| Al$_4$Si$_4$Zn | 551.7 | 546.9 | 554.3 | 543.8 | 541.7 | 546.1 |
| Al$_5$Si$_4$Zn | 552.4 | 548.4 | 553.3 | 546.2 | 544.2 | 545.5 |
| Al$_6$Si$_4$Zn | 550.3 | 547.0 | 550.1 | 545.4 | 540.4 | 542.1 |
| Al$_7$Si$_4$Zn | 549.4 | 546.6 | 548.7 | 541.4 | 536.8 | 543.3 |
| | Liquidus Temperature (°C) | | | | | |
| Al$_4$Si$_4$Zn | 631.8 | 641.3 | 651.7 | 664.7 | 686.9 | 714.9 |
| Al$_5$Si$_4$Zn | 625.5 | 633.0 | 643.0 | 654.1 | 681.3 | 701.4 |
| Al$_6$Si$_4$Zn | 620.0 | 625.3 | 636.1 | 651.2 | 673.0 | 694.3 |
| Al$_7$Si$_4$Zn | 612.9 | 615.7 | 621.4 | 642.3 | 663.9 | 682.7 |
| | Semisolid interval (°C) | | | | | |
| Al$_4$Si$_4$Zn | 80.1 | 94.4 | 97.4 | 120.9 | 145.2 | 168.8 |
| Al$_5$Si$_4$Zn | 73.1 | 84.6 | 89.7 | 107.9 | 137.1 | 155.9 |
| Al$_6$Si$_4$Zn | 69.7 | 78.3 | 86.0 | 105.8 | 132.6 | 152.2 |
| Al$_7$Si$_4$Zn | 63.5 | 69.1 | 72.7 | 100.9 | 127.1 | 139.4 |

**Table 5.** Comparison of the variables area between DSC curve and baseline (*A* in W°C/g), enthalpy change (Δ*H* in V/g), solidus (*S* in °C), liquidus (*L* in °C) and semisolid interval (*S–L* in °C) for extreme (lower and upper) values of Si content, heating rate and sample mass.

| Condition | Ratio [4] | | | | |
|---|---|---|---|---|---|
| (Lower–Upper) | *A* | Δ*H* | *S* | *L* | *S-L* |
| Si (4–7 wt.%) | 1.01 | 0.24 | 0.99 | 0.96 | 0.80 |
| *dT/dt* (5–25 °C/min) | 5.04 | 0.93 | 0.98 | 1.07 | 1.66 |
| mass (20–200 mg) | 0.76 | 0.77 | 1.00 | 1.05 | 1.31 |

[4] Ratio between two extreme (lower and upper) conditions, e.g., for the area between the DSC curve and baseline the ratio between Si contents is given by $A_{7Si}/A_{4Si}$; between heating rates, the ratio is given by $A_{25°C/min}/A_{5°C/min}$; between sample masses, by $A_{200mg}/A_{20mg}$; and so on for the other parameters.

### 3.1. Effect of Si Content

As the Si content increases from 4 to 7 (top to bottom in Figure 1), the valley corresponding to the eutectic transformation becomes progressively deeper on the *HFR* axis and more widely spread along the temperature axis. Both behaviors are due to the larger amount of eutectic being formed (the eutectic composition for the system studied is 11.8 wt.%Si according to the CALPHAD simulation at equilibrium). For the valley corresponding to the Al-α phase, the behavior is the opposite: the valley becomes progressively smaller on the *HFR* axis and narrower on the temperature axis. These trends can be seen in detail in Figure 2, in which the alloys with 4 and 7 wt.%Si are compared for the same heating rate and sample mass.

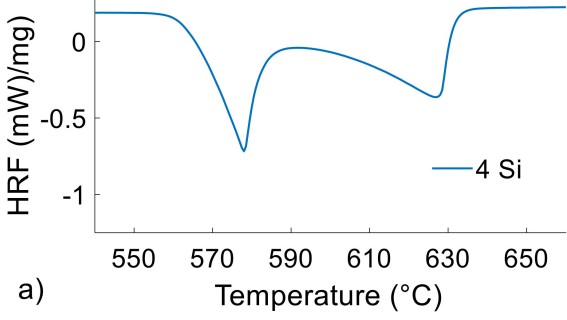 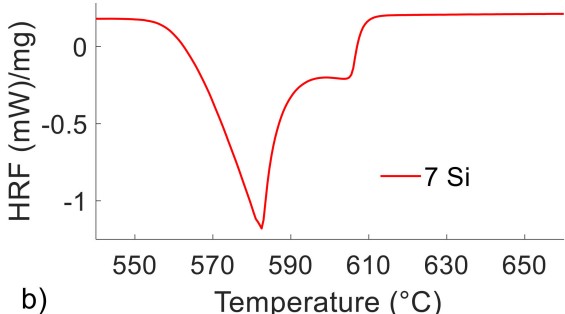

**Figure 2.** DSC heat flow rate (*y*-axis) vs. temperature (*x*-axis) curves during melting of the $Al_4Si_4Zn$ (**a**) and $Al_7Si_4Zn$ (**b**) alloys at 5 °C/min with a DSC sample mass of 90 mg.

The area between the DSC curves and baseline is barely affected by Si content (for a given sample mass and heating rate), varying around only 1% when the Si content is raised from 4 to 7 wt.% (ratio $A_{7Si}/A_{4Si}$ = 1.01 in Table 5). The enthalpy changes and the semisolid interval decreases as the Si content increases, and the average reductions are 76% and 20%, respectively, as the eutectic composition is approached (Table 5). This decrease is due more to a reduction in the liquidus temperature (average reduction of 4%, see Table 5) than to an increase in the solidus (maximum change of 1%, see Table 5), which is in agreement with the nature of the eutectic reaction (constant solidus and rapidly decreasing liquidus as the eutectic composition is approached).

*3.2. Effect of Heating Rate*

As the heating rate is increased from 5 to 25 °C/min (blue to green curves in the graphs in Figure 1, the intensity of the valleys increases steadily because of the larger heat flow needed for the phase transformations to occur in progressively shorter times. In addition, the curves are spread over a wider temperature range (i.e., the semisolid interval becomes larger) and the transformations are delayed (pushed to higher temperatures). The semisolid interval increases on average 66% when the heating rate increases from 5 to 25 °C/min (Table 5) for a given sample mass and Si content. This increase is due more to a rise in the liquidus (average increase of 7%) than to a reduction in the solidus (change of only 2%). This behavior is the result of the combined effect of two factors: (1) higher heating rates and (2) progressively higher temperatures as the reaction progresses; together, these increase the reaction activation energy (E) compared with cycles performed near-equilibrium and lower-temperature transformations [19]. An increase in E results in a larger thermal power requirement, causing the reaction to occur at a progressively higher temperature [20], which in turn leads to a cumulative effect at the end of the melting transformation (increase in the liquidus). These trends can be seen in detail in Figure 3, in which the heating rates of 5 and 25 °C/min are compared for the same Si content and sample mass.

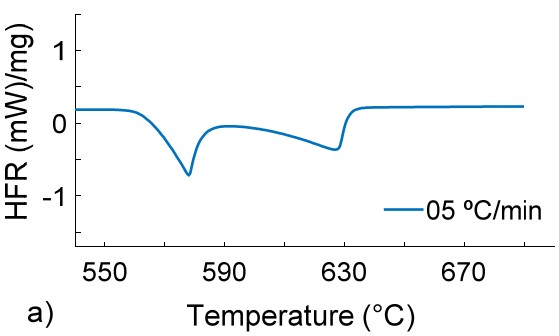 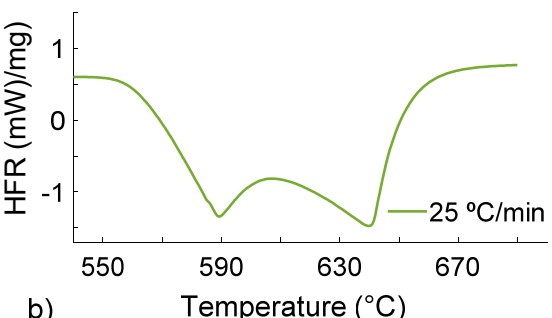

**Figure 3.** DSC heat flow rate (*y*-axis) vs. temperature (*x*-axis) curves during melting of the $Al_4Si_4Zn$ alloy at 5 (**a**) and 25 (**b**) °C/min with a DSC sample mass of 90 mg.

The area between the DSC curves and baseline increases with the increase in the heating rate. The ratio between the areas corresponding to 5 and 25 °C/min (for a given Si content and sample mass) is approximately $A_{25}/A_5 = 5$ for all the conditions tested (Table 5), which is exactly the ratio between these heating rates. This value can be explained as follows: as the enthalpy change is the integral of the DSC output (thermal power) with respect to time, for the heating rates of 25 and 5 °C/min the times ($t$) involved in the melting follow the proportion $t_5 = 5t_{25}$ (subscripts referring to the heating rates). Since the enthalpy change should be the same for both heating rates given a constant sample mass (same amount of matter being transformed), the following equation is assumed:

$$\int_0^{t_5} P_5 \, dt = \int_0^{t_{25}} P_{25} \, dt, \tag{1}$$

where $P$ is the thermal power (energy per unit time) and $t$ is the time for the melting transformation to take place (the subscripts 5 and 25 refers to the heating rates). As can be seen from Equation (1), $P_{25} = 5P_5$. When the thermal power is integrated with respect to temperature (the areas between the DSC curves and baseline), i.e.,

$$\int_{S_5}^{L_5} P_5 \, dT = P_5 \, \Delta T_{SL-5}, \tag{2}$$

and

$$\int_{S_{25}}^{L_{25}} 5P_{25} \, dT = P_{25} \, \Delta T_{SL-25}, \tag{3}$$

It can be seen that the areas between the DSC curves and baseline are linearly dependent on the heating rate and on the length of the semisolid interval:

$$A_5 = P_5 \, \Delta T_{SL-5}, \tag{4}$$

and

$$A_{25} = 5P_5 \, \Delta T_{SL-25}, \tag{5}$$

where $L$ and $S$ are the solidus and liquidus and $\Delta T_{SL}$ is the semisolid interval in Equations (2)–(5). Considering an identical semisolid interval for both conditions, the area is totally dependent on the heating rate, and the equations fit the data in Table 5 ($A_{25}/A_5 = 5$).

However, $\Delta T_{SL}$ is not invariant with the increase in heating rate. As seen before, it increases by on average 66% from 5 to 25 °C/min (Table 5), so the only possible explanation for the ratio $A_{25}/A_5 = 5$ is that the enthalpy change decreases with the increase in heating rate, compensating for the increase in $\Delta T_{SL}$. This is exactly what is observed empirically: the enthalpy change, $\Delta H$, corresponding to the melting reaction, is reduced by 7% on average (Table 5) with the increase in the heating rate.

The decrease in the measured $\Delta H$ with the increase in heating rate (for identical masses) can be explained by a difference in the heat loss when the heating rate is changed. The samples subjected to longer cycles will be exposed to the environment (DSC crucible and furnace) for longer periods, resulting in larger heat loss to complete the same state transformation and vice versa for shorter cycles. The extra heat supplied by the equipment to compensate for this heat loss is not taken into account in the theoretical relationship between $\Delta H$ and $dT/dt$ but is substantial in real cycles (7% of the power supplied, as seen before) and mistakenly appears in the DSC curves as if this power were used for the transformation itself.

The height ($h$) [21], shape and position [22] of each individual *HFR* valley with respect to the curve baseline are affected by the change in enthalpy ($\Delta H$), the heating rate ($dT/dt$),

the heat capacity of the sample (CS) and the thermal resistance ($R'$) of the DSC apparatus. The relationship between the height [21] and these parameters is given by:

$$h = -\left(\frac{dT}{dt}\right) \times C_S + \left[\left(\frac{dT}{dt}\right)2 \times C_S^2 + 2 \times \left(\frac{dT}{dt}\right) \times \frac{\Delta H}{R}\right]^{\frac{1}{2}},$$ (6)

It follows that the larger the heating rate, the deeper the valley: exactly as seen when the blue and green curves in Figure 1 are compared. Ideally, the baselines of the DSC curves should be around zero. However, because of mismatching of the thermal properties of the sample and reference materials, and possible asymmetry in the construction of the sample and reference holders, the baselines are displaced from the reference position. This displacement ($h'$) is proportional to the calibration factor ($K$), heating rate ($dT/dt$) and heat capacity of the sample ($C_S$); the relationship between these parameters is given by the following expression [19]:

$$h' = K \times \left(\frac{dT}{dt}\right) \times C_S,$$ (7)

Hence, higher heating rates produce more marked displacements in the baseline, as seen in the curves in Figure 1. Furthermore, sloping baselines can be generated as a result of differences in the emissivity of the reference and crucible materials and differences in the $C_S$ between the low-temperature form (solid) and high-temperature form (liquid) of the sample material [19,23,24].

Another feature observed with the increase in the heating rate is the overlapping of the valleys, making identification of the transition between the Si-eutectic and the Al-$\alpha$ phase transformations difficult. This effect, which is undesirable from the point of view of thixoformability analysis, is due to the increase in the temperature gradients in the sample as a result of the increase in heating rate, i.e., the heterogeneous heat transfer flow along the sample leads to the same phase transformation occurring at different locations and times inside the same sample. The sum of the contributions of each "portion" of phase transformation leads to a result similar to that for a diffuse transition. This same hypothesis can be used to justify the larger spread of the curves along the temperature axis with the increase in the heating rate discussed earlier.

*3.3. Effect of the Sample Mass*

Progressive overlapping of the DSC valleys is observed with the increase in the sample mass (left to right in Figure 1), making identification of the transition between the two phase transformations difficult. Additionally, a 31% increase in the semisolid interval can be observed (for constant Si content and heating rate) as a consequence of a 5% increase in the liquidus temperature. The solidus temperature is practically unaffected by the increase in the sample mass (ratio $S_{200mg}/S_{20mg} = 1$). These trends can be attributed to the larger temperature gradient generated in the heavier samples, resulting in temporally and spatially more heterogeneous melting along the sample. The overall result is seen as delayed, overlapping transformations in the DSC curves. These trends can be seen in detail in Figure 4, in which the masses of 20 and 200 mg are compared for the same Si content and heating rate.

Another effect seen in the DSC curves is the influence of the sample mass on the slope of the curve at the beginning of the melting reaction, i.e., the angle ($\theta$) between the baseline and the DSC curve [13]. This angle is indicated in the curves in Figure 1d–f for reference. The values of $\theta$ measured on the curves were, regardless of the Si content and heating rate used, $\theta_{20} = 100.5°$, $\theta_{90} = 118.0°$ and $\theta_{200} = 130.0°$. The respective $R$ values were $R_{20} = 0.18$, $R_{90} = 0.53$ and $R_{200} = 0.85$ (note that $R = 1/\tan(\theta)$). As can be seen, $R_{200} > R_{90} > R_{20}$, which is in agreement with the fact that a larger sample offers greater resistance to heat transfer along its volume. When the increases in mass, $m_{200/20} = 10$, $m_{90/20} = 4.5$ and $m_{200/90} = 2.2$, and the respective increases in thermal resistance, $R_{200/20} = 4.7$, $R_{90/20} = 2.9$ and $R_{200/90} = 1.6$, are compared, it can be concluded that the resistance $R$ increases in

proportion to $m^{0.7}$. This result is in agreement with the fact that any change in $R$ is due to the sample (change in mass) since both instrument and crucible are the same for all the cycles. As a practical outcome, this result shows that a lower sample mass is once again recommended to enhance the precision of the thermoanalytical measurement since this results in a steeper initial slope on the DSC curve associated with the phase transformations.

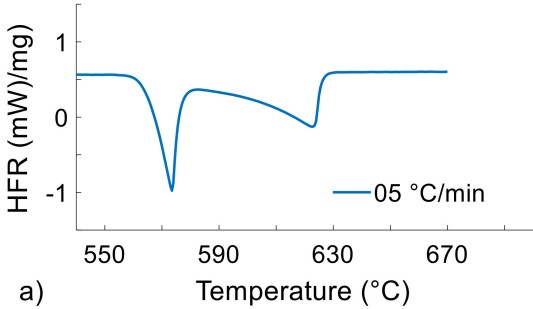 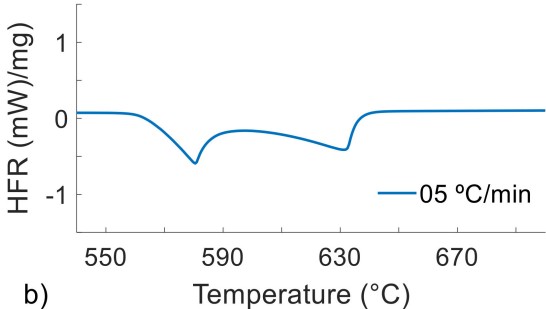

a)  b)

**Figure 4.** DSC heat flow rate (*y*-axis) vs. temperature (*x*-axis) curves during melting of the Al$_4$Si$_4$Zn alloy at 5 °C/min with DSC sample masses of 20 (**a**) and 200 (**b**) mg.

Although the values of the area between the DSC curves and baseline and the enthalpy change in Tables 2 and 3 are already divided by the sample mass, reductions of, on average, 24% and 23% are observed for $A$ and $\Delta H$, respectively, with the increase in the sample mass (for constant Si content and heating rate, see Table 5). This result can be explained by the different heat loss when different sized samples are used. Table 6 shows the average dimensions of the samples weighing 20, 90 and 200 mg. The ratio between the surface area and volume ($A_s/V$) of the samples decreases by 50% when the sample mass increases from 20 to 200 mg. The heat that is not used to produce the phase transformations is transferred from the sample to the environment, mainly by forced convection to the furnace atmosphere (heat loss through the flat bottom of the sample can be ignored because of the good thermal insulation of the crucible material). With the decrease in the $A_s/V$ ratio, there is a significant reduction in heat loss compared with the total heat used by the DSC device. Thus, in contrast to all the results discussed above, the enthalpy change can be more precisely measured for heavier samples, since this parameter is the total heat exchange during the transformation and therefore is greatly affected by the heat loss during the test.

**Table 6.** Average DSC sample dimensions.

| Mass (mg) | Height (mm) | Radius (mm) | $A_s$ [5] (mm$^2$) | $V$ (mm$^3$) | $A_s/V$ (mm$^{-1}$) | Ratio (-) |
|---|---|---|---|---|---|---|
| 20 | 1.0 | 3.6 | 64.06 | 41.28 | 1.55 | 1.00 |
| 90 | 2.1 | 4.2 | 108.71 | 112.54 | 0.96 | 0.62 |
| 200 | 2.6 | 5.1 | 164.49 | 212.18 | 0.77 | 0.50 |

[5] The surface area was considered to be the effective area where a significant amount of heat exchange with the environment occurred (i.e., the flat top and the side of the cylindrical samples).

Note that the assumptions made so far ignore possible sources of error in the DSC operation: the assumption of a constant heating rate for the sample, which fails to take into account the effect of self-cooling or self-heating during reactions [25], and the assumption that the temperatures of the reference and sample are identical to those predicted by the heating program (in real cycles there may be a considerable lag) [24,26], among several other factors [27–36].

*3.4. Thixoforming Working Window*

In order to understand how the trends observed in the DSC curves affect the thixoforming working window ($\Delta T_{THIXO}$) [6], the analysis applied to the DSC data is shown schematically in Figure 5 in the following order: starting with the original *HFR* curve (black curve in Figure 5a), the derivative of the *HFR* curve with respect to *T* is plotted

(red curve in Figure 5a) and the temperatures of the beginning ($T_{SSML}$) and end ($T_{SSMH}$) of $\Delta T_{THIXO}$ are determined using a baseline = 0 established by the differentiation method (DM) [10]. Using the solidus and liquidus temperatures (also obtained by the DM), the limits of integration of partial areas between the DSC curve and the baseline are set and the $f_S$ vs. $T$ curve (blue curve in Figure 5b) is drawn [12]. The positions on the $f_S$ curve corresponding to $T_{SSML}$ and $T_{SSMH}$ on the $dHFR/dT$ vs. $T$ curve provide the solid fractions corresponding to these temperatures, i.e., $fs_{SSML}$ and $fs_{SSMH}$ (Figure 5c). Finally, the derivative of $f_S$ with respect to temperature, i.e., the $df_S/dT$ vs. $T$ (or sensitivity curve), is obtained (light blue curve in Figure 5d). By examining the $df_S/dT$ values within $\Delta T_{THIXO}$, it can be established whether the $df_S/dT_{(SSML-SSMH)} < 0.03\ °C^{-1}$ criterion is met [1], thereby guaranteeing adequate process control and stability.

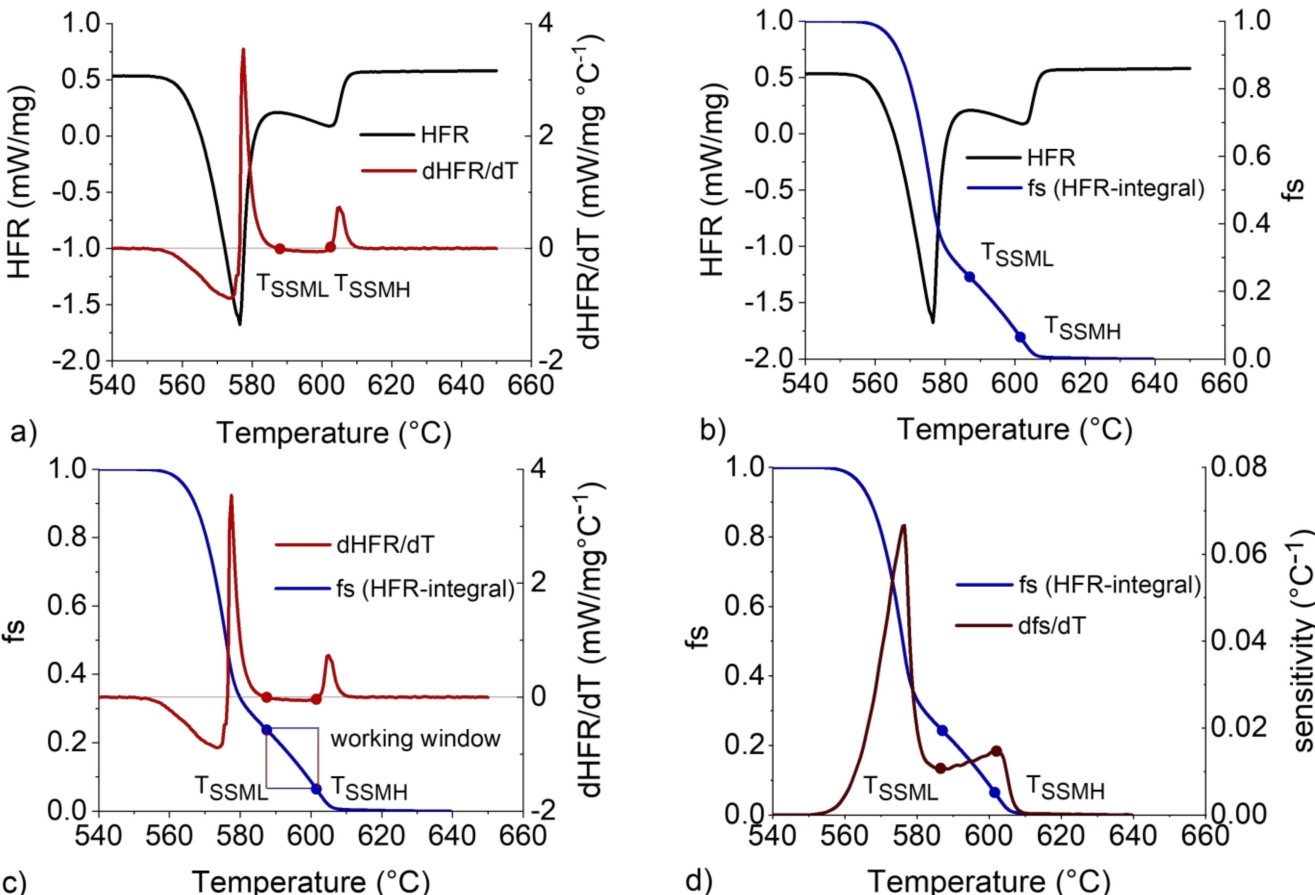

**Figure 5.** *HFR* vs. *T* (black), *dHFR/dT* vs. *T* (red), *fs* vs. *T* (dark blue) and *dfs/dT* vs. *T* (light blue) curves for $Al_7Si_4Zn$ alloy with a 5 °C/min heating rate and 20 mg sample mass; the temperatures/solid fractions at the beginning (*SSML*) and end (*SSMH*) of the thixoforming processing working window are shown as solid dots. (**a**) Firt step-obtaining of the *dFHR/dT/dT* from *FHR*, (**b**) Second step-Partial integration of the *HFR* curve to obtain the *fs* × *T* curve, (**c**) derivative curve of the solid fraction as equivalent of the sensitivity, (**d**) solid fraction and its corresponding sensitivity as a function of temperature.

To identify $\Delta T_{THIXO}$ properly by the DM it is crucial that both the main transformations (Al-$\alpha$ and Si-eutectic) produce well-defined valleys in the DSC curves, i.e., that an inversion from a positive to a negative curvature be observed at the Al-$\alpha$/Si-eutectic interface ($X_1$, in Figure 6a). This is seen as the "peak", formed between the two valleys in the curves for 4–6 wt.%Si. If this curvature inversion is not observed $X_2$, as in the curve for 7 wt.%Si (red curve in Figure 6b), then $T_{SSML}$ and $T_{SSMH}$ cannot be identified, since the absence of

a curvature inversion will result in a derivative curve that does not cross the baseline = 0 used in the method.

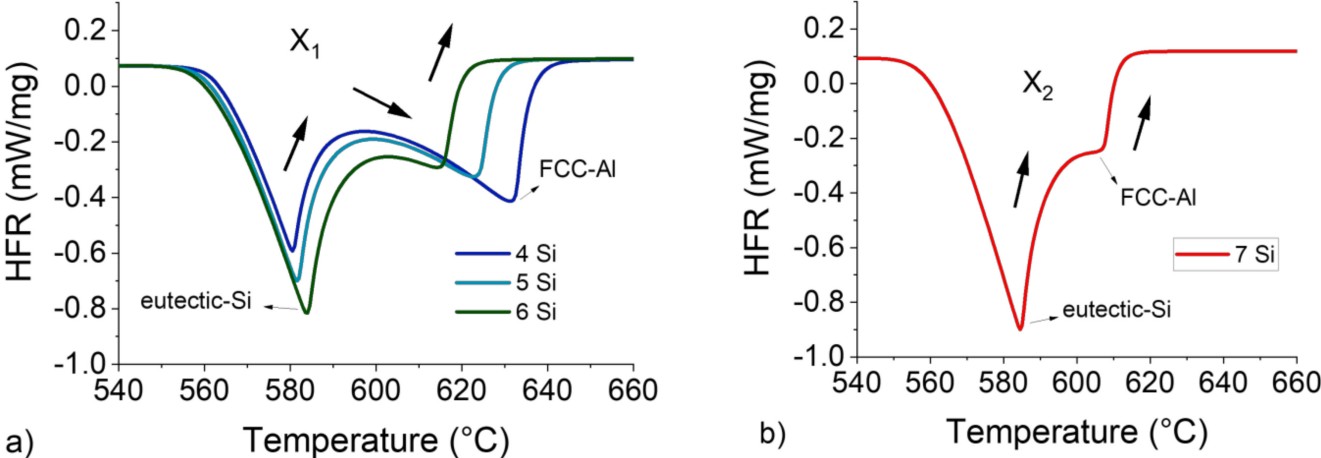

**Figure 6.** *HFR* vs. *T* curves for 4 to 6 wt.%Si with a 5 °C/min heating rate and 200 mg sample mass, complete curvature inversion at the Al-$\alpha$/Si-eutectic interface $X_1$ (**a**), and *HFR* vs. *T* curve for 7 wt.%Si with a 5 °C/min heating rate and 200 mg sample mass, with incomplete curvature $X_2$ (**b**).

This situation is illustrated in Figure 7, where the DSC derivative corresponding to 7 wt.%Si ($X_2$ in Figure 7b) does not cross the baseline and $\Delta T_{THIXO}$ cannot be defined. For the other alloys (4–6 wt.%Si), the derivatives cross the baseline and both $T_{SSML}$ and $T_{SSMH}$ can be determined. (Note that the two points where the curve crosses the baseline are identified as $T_{SSML}$ and $T_{SSMH}$ in the DM).

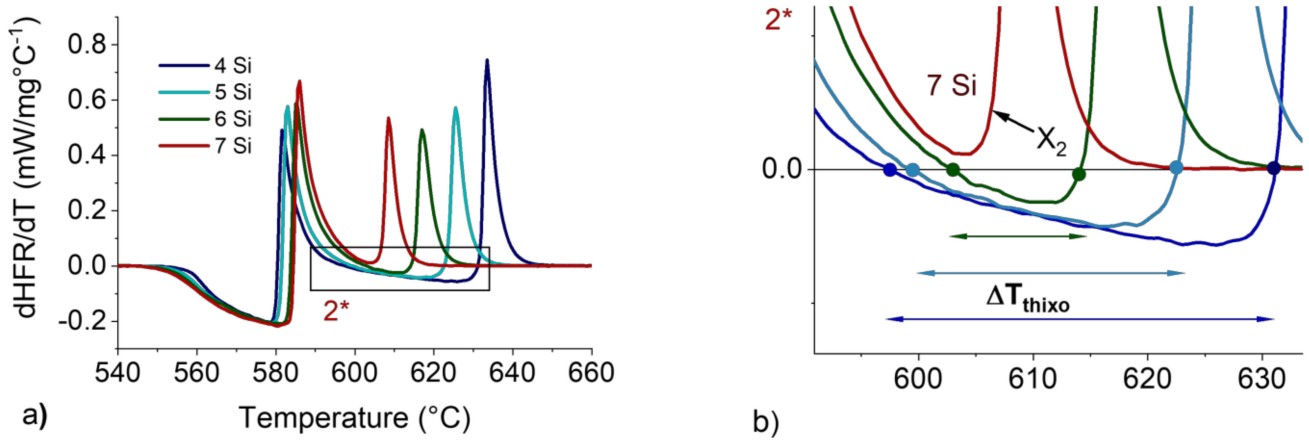

**Figure 7.** *dHFR/dT* vs. *T* curves for 4 to 7 wt.%Si with a 5 °C/min heating rate and 200 mg sample mass (**a**). Magnification at the $\Delta T_{THIXO}$ zone (**b**).

As can be seen in Figure 7b, the fact that $\Delta T_{THIXO}$ cannot be identified for 7 wt.%Si is a direct consequence of the tendency for the working window to decrease as the Si content increases. With progressive additions of Si, both $T_{SSML}$ and $T_{SSMH}$, particularly the latter, are displaced to higher and lower temperatures, respectively, bringing them closer to each other and narrowing the working window. As they are further displaced, the two critical points will eventually become a single point on the curve where it touches the baseline (in which case only $T_{SSML}$, i.e., this point, can be identified); finally, if the curve does not touch the baseline anymore, neither $T_{SSML}$ nor $T_{SSMH}$ can be identified, as seen for 7 wt.%Si. The narrowing of the working window with increasing Si content is analyzed in detail in a previous work [11].

In order to identify $\Delta T_{THIXO}$ for 7 wt.%Si, cycles with other heating rates were tested for this alloy. The DSC derivatives for cycles with 5 (as in Figure 7), 10, 15, 20 and 25 °C/min heating rates are shown in Figure 8, from which it can be seen that the problem worsens as the heating rate increases since the curves are displaced farther from the baseline. In addition, as the heating rate increases, the curves are pushed to higher temperatures. This morphological behavior, i.e., the delay and enlargement of the curves as the distance from equilibrium increases, as discussed earlier in this work, is inherited from the original DSC curves.

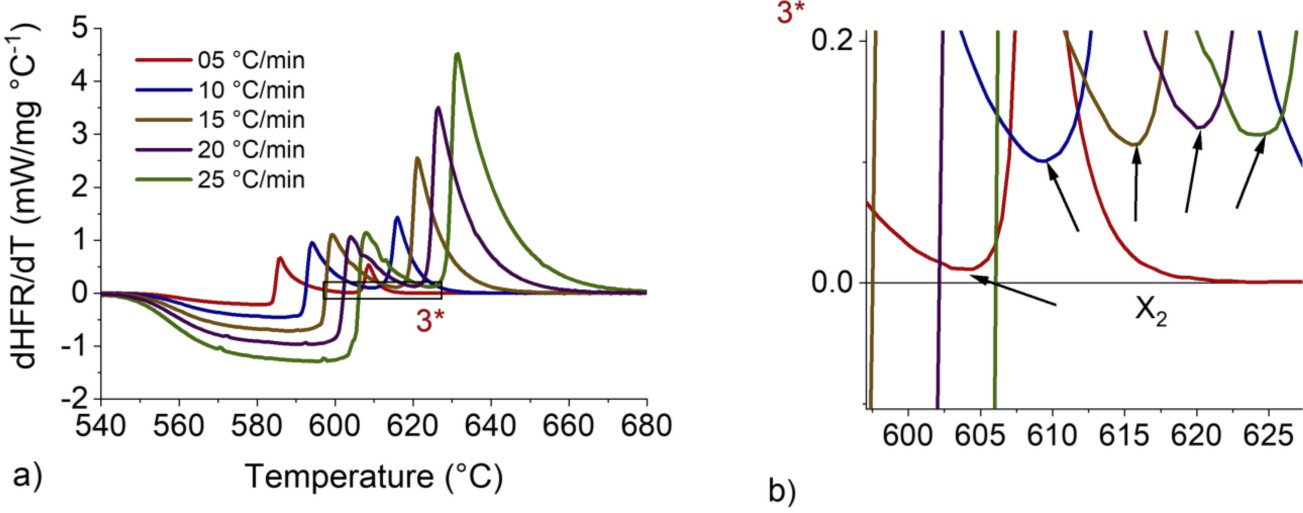

**Figure 8.** *dHFR/dT* vs. *T* curve for 7 wt.%Si with heating rates from 5 to 25 °C/min and sample mass of 200 mg (**a**). Magnification at the $\Delta T_{THIXO}$ zone (**b**).

A further attempt to determine $\Delta T_{THIXO}$ was made by reducing the DSC sample mass from 200 mg (Figures 5 and 7) to 90 mg (Figure 9) and 20 mg (Figure 10). This resulted in the valleys becoming steeper and the derivatives crossing the zero baseline more often, facilitating identification of the critical temperatures and overcoming the limitation observed before for 200 mg samples (Figure 8).

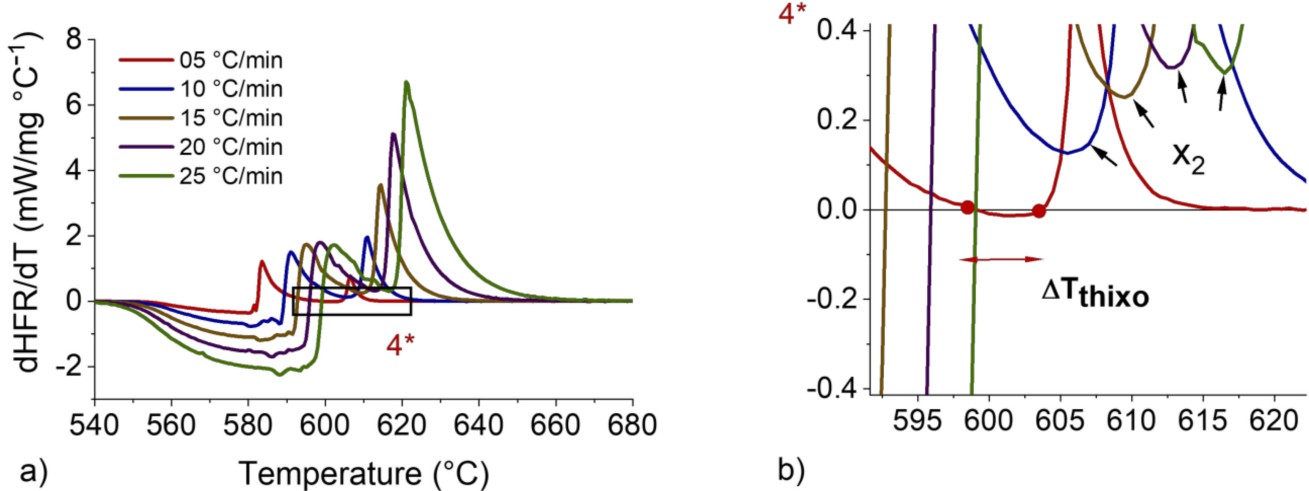

**Figure 9.** *dHFR/dT* vs. *T* curve for 7 wt.%Si with heating rates from 5 to 25 °C/min and sample mass of 90 mg (**a**). Magnification at the $\Delta T_{THIXO}$ zone (**b**).

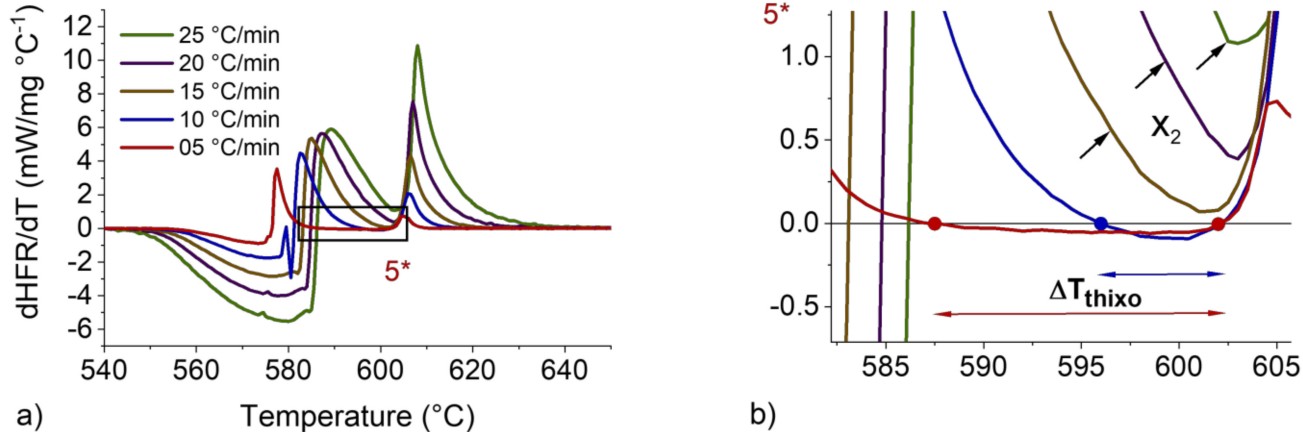

**Figure 10.** *dHFR/dT* vs. *T* curve for 7 wt.%Si with heating rates from 5 to 25 °C/min and sample mass of 20 mg (**a**). Magnification at the $\Delta T_{THIXO}$ zone (**b**).

Reduction in the sample mass to 90 mg allowed $\Delta T_{THIXO}$ for the 7 wt.%Si alloy heated at 5 °C/min to be identified, and a further reduction (to 20 mg) allowed the interval to be identified not only at 5 °C/min but also at 10 °C/min. With the other heating rates $\Delta T_{THIXO}$ remained unidentifiable for this alloy. As a general trend, it can be concluded that the lower the heating rate, Si content and sample mass, the easier it is to identify $\Delta T_{THIXO}$ for the range of alloys and conditions studied here, as shown schematically in Figure 11.

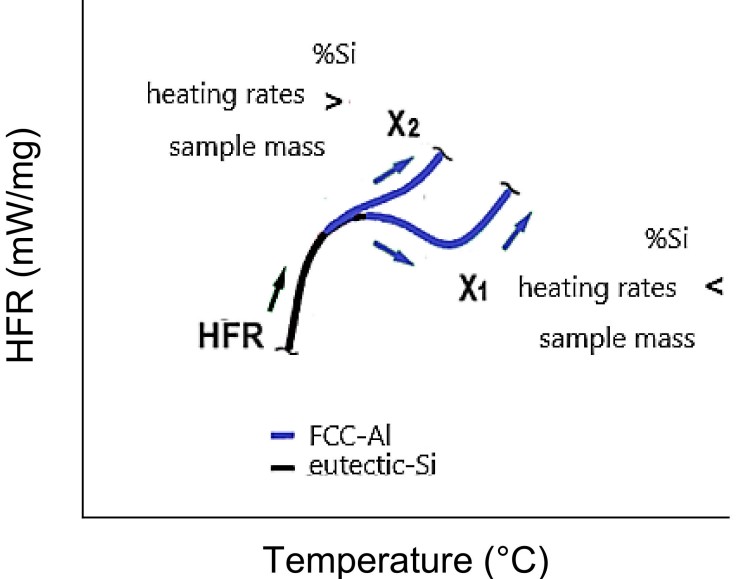

**Figure 11.** Correlation between Si content, heating rate and DSC sample mass and the feasibility of identifying $\Delta T_{THIXO}$ for the Al-Si-Zn system.

The $f_S$ vs. *T* curves obtained by DSC for the Al-(4-7)Si-4Zn alloys at the lowest and highest heating rates (5 and 25 °C/min, respectively) are shown in Figure 12 for sample masses of 200 (a), 90 (b) and 20 mg (c). The corresponding processing windows, $\Delta T_{THIXO}$ is shown in Table 7. As discussed before, as the sample mass decreases, the critical temperatures for a progressively larger number of cycles (heating rates) and alloys (Si contents) can be identified.

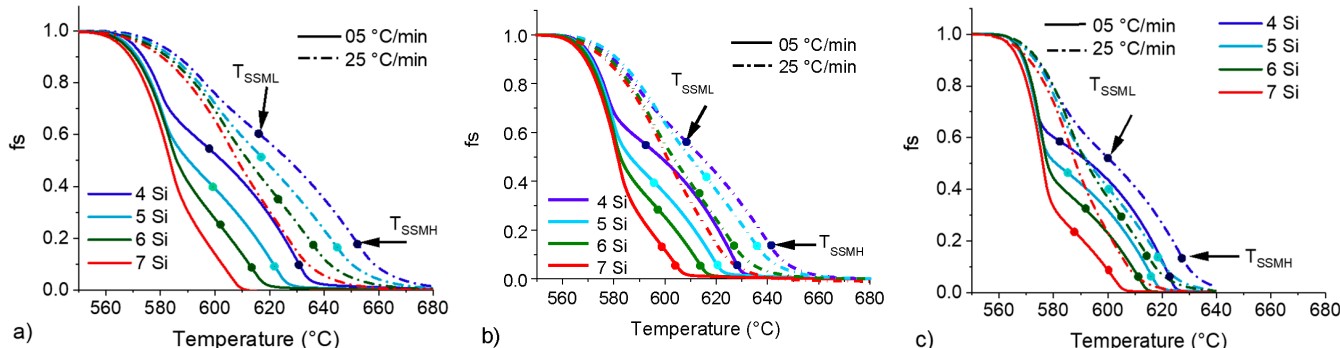

**Figure 12.** $f_S$ vs. $T$ curve for 4, 5, 6 and 7 wt.%Si with heating rates of 5 and 25 °C/min and sample masses of 200 (**a**) 90 (**b**) and 20 mg (**c**).

**Table 7.** The $\Delta T_{THIXO}$ $Al_XSi_4Zn$ for 05 °C/min and 25 °C/min, with a sample mass 200 mg, 90 mg, and 20 mg.

| Mass (mg) | wt.%Si | $\Delta T_{THIXO}$ (°C) | |
|---|---|---|---|
| | | **05 °C/min** | **25 °C/min** |
| 200 | 4Si | 33.8 | 38.1 |
| | 5Si | 23.3 | 27.7 |
| | 6Si | 10.9 | 13.5 |
| | 7Si | NI | NI |
| 90 | 4Si | 35 | 32.3 |
| | 5Si | 23.5 | 18.2 |
| | 6Si | 16.7 | 12.7 |
| | 7Si | 4.6 | NI |
| 20 | 4Si | 39.8 | 26.8 |
| | 5Si | 31.4 | 17.4 |
| | 6Si | 24.7 | 9.9 |
| | 7Si | 14.8 | NI |

Regardless of the sample mass used, an increase in Si content reduces $\Delta T_{THIXO}$ as this is above the eutectic transformation (the larger the eutectic field, the smaller the remaining Al-$\alpha$ phase temperature range for thixoforming). On the other hand, the effect of the heating rate on $\Delta T_{THIXO}$ is directly influenced by the DSC sample mass: for 200 mg, the increase in the heating rate leads to enlargement of $\Delta T_{THIXO}$ while for 90 and 20 mg this trend is inverted. It has already been discussed in connection with Figure 1 that an increase in heating rate leads to progressive overlapping of the Si-eutectic and Al-$\alpha$ phase valleys. This effect leads to a progressively narrower $\Delta T_{THIXO}$ until the critical temperatures can no longer be identified [11], exactly as observed for 20 and 90 mg samples. For 200 mg samples, however, the heat transfer in the sample becomes more heterogeneous when the heating rate is increased. A more critical time-dependent heat flow response to the power supplied leads to an even more "diffuse" bulk transformation. The result is a curve, and $\Delta T_{THIXO}$, with a greater spread along the temperature axis.

### 3.5. Lower Limit of the Semisolid Window

The $f_S$ vs. $T$ curves obtained by CALPHAD simulation at the near-equilibrium Scheil condition for 4 to 7 wt.%Si are shown in Figure 13 (solid lines) along with the experimental curves for heating rates of 5 and 25 °C/min (dashed and dotted–dashed lines, respectively) using 200 (a), 90 (b) and 20 mg (c) DSC samples. The temperature corresponding to the lower limit of the working window ($T_{SSML}$) is shown in Table 8. Special attention is given to $T_{SSML}$ in Figure 13 because in thixoforming practice a knowledge of this point is enough to determine the target operating temperature, which is usually chosen to be a few degrees above $T_{SSML}$ in order to save energy and guarantee the structural stability of the semisolid slurry immediately before thixoforming. The temperatures corresponding to the upper

limit of the working window $T_{SSMH}$ identified here for the Al-(4-7)Si-5Zn system for all the studied conditions correspond to $f_S < 0.2$ (Figure 12), an extremely low solid fraction of little practical use.

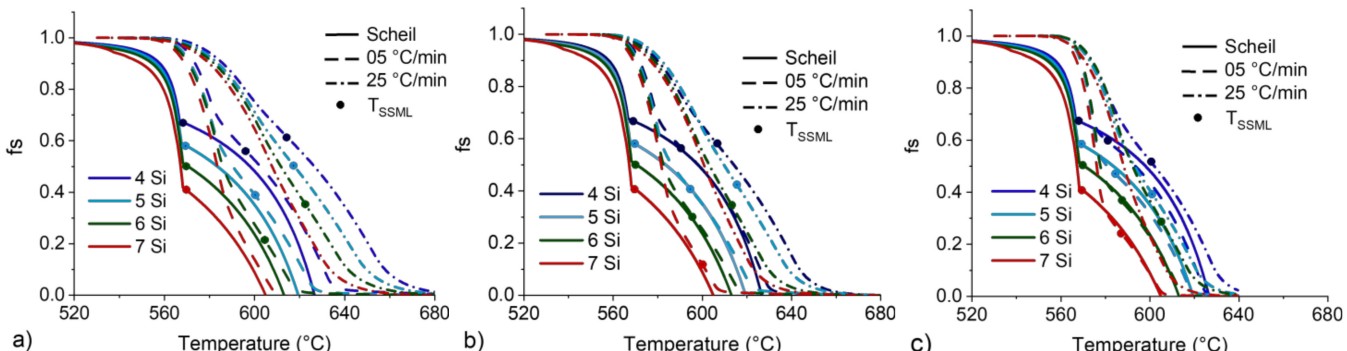

**Figure 13.** $f_S$ vs. $T$ curves for 4 to 7 wt.%Si from CALPHAD simulation at the Scheil condition and DSC curves for heating rates of 5 and 25 °C/min and 200 (**a**), 90 (**b**) and 20 mg (**c**) DSC samples.

**Table 8.** The $T_{SSML}$ for Al$_X$Si$_4$Zn in the Scheil condition, 05 °C/min and 25 °C/min, with sample mass 200 mg, 90 mg and 20 mg.

| Mass (mg) | wt.%Si | $T_{SSML}$ (°C) | | |
|---|---|---|---|---|
| | | Scheil | 05 °C/min | 25 °C/min |
| 200 | 4Si | 566 | 597.5 | 614.0 |
| | 5Si | 567 | 599.0 | 616.0 |
| | 6Si | 568 | 603.0 | 621.6 |
| | 7Si | 568 | NI | NI |
| 90 | 4Si | 566 | 591.6 | 607.2 |
| | 5Si | 567 | 595.3 | 615.4 |
| | 6Si | 568 | 593.5 | 609.4 |
| | 7Si | 568 | 599.2 | NI |
| 20 | 4Si | 566 | 582.4 | 600.1 |
| | 5Si | 567 | 584.3 | 602.2 |
| | 6Si | 568 | 586.2 | 604.8 |
| | 7Si | 568 | 587.8 | NI |

The increase in Si content leads to the already expected increase in $T_{SSML}$ due to the reduction in the Al-$\alpha$ field with the formation of a larger amount of eutectic. Additionally, the displacement of $T_{SSML}$ to higher temperatures with the progressive distancing from equilibrium is expected as a result of the delay and spreading of the phase transformations with the increase in the heating rate.

When the DSC sample mass is reduced, $T_{SSML}$ at 5 °C/min and $T_{SSML}$ simulated at the Scheil condition are similar. The largest difference observed between the Scheil condition and 5 °C/min (Table 8) was 35 °C (for 6 wt.%Si and 200 mg), which reduced to only 16.4 °C (for 4 wt.%Si and 20 mg). This can be attributed to more accurate measurement (yielding results which are in better agreement with the simulation) for lighter masses, which are not severely affected by heat transfer heterogeneity, as discussed earlier. It should be kept in mind that CALPHAD simulates the semisolid transformation during solidification, so the differences observed are due not only to the distancing from equilibrium but also to the inversion in the heat flow considered as in DSC the melting reaction is measured.

When the experimental cycles at 5 and 25 °C/min are compared, the variations in $T_{SSML}$ with heating rate are far smaller, ranging only from 20.1 °C to 15.6 °C (for 5 and 4 wt.%Si, respectively, both observed for 90 mg). This result may indicate that the effect of the heating rate in delaying and extending the transformations along the temperature axis for 25 °C/min is sufficient in itself for the effects of the sample mass on the transformations to be negligible.

*3.6. CALPHAD vs. DSC*

A comparison between the simulated (Scheil) and experimental sensitivity curves for all the alloys and sample masses studied is shown in Figure 14 for heating rates of 5 °C/min (left) and 25 °C/min (right). As these curves inherit the morphology of their respective $f_S$ vs. $T$ curves, the lower the Si content, sample mass and heating rate, the steeper the sensitivity peaks. It can be seen that the lower heating rate curves (5 °C/min, left) are closer to the Scheil curves than the higher rate curves (25 °C/min, right) are for each alloy and sample mass; this is as expected as the former are nearer equilibrium than the latter. The greatest similarity between the simulation and experimental data was achieved with the lowest heating rate and mass (Figure 14e) independently of Si content. For all the experimental curves, $\Delta T_{THIXO}$ determined by the DM was maintained because the sensitivity within this interval is below 0.03 °C$^{-1}$ for all the cycles.

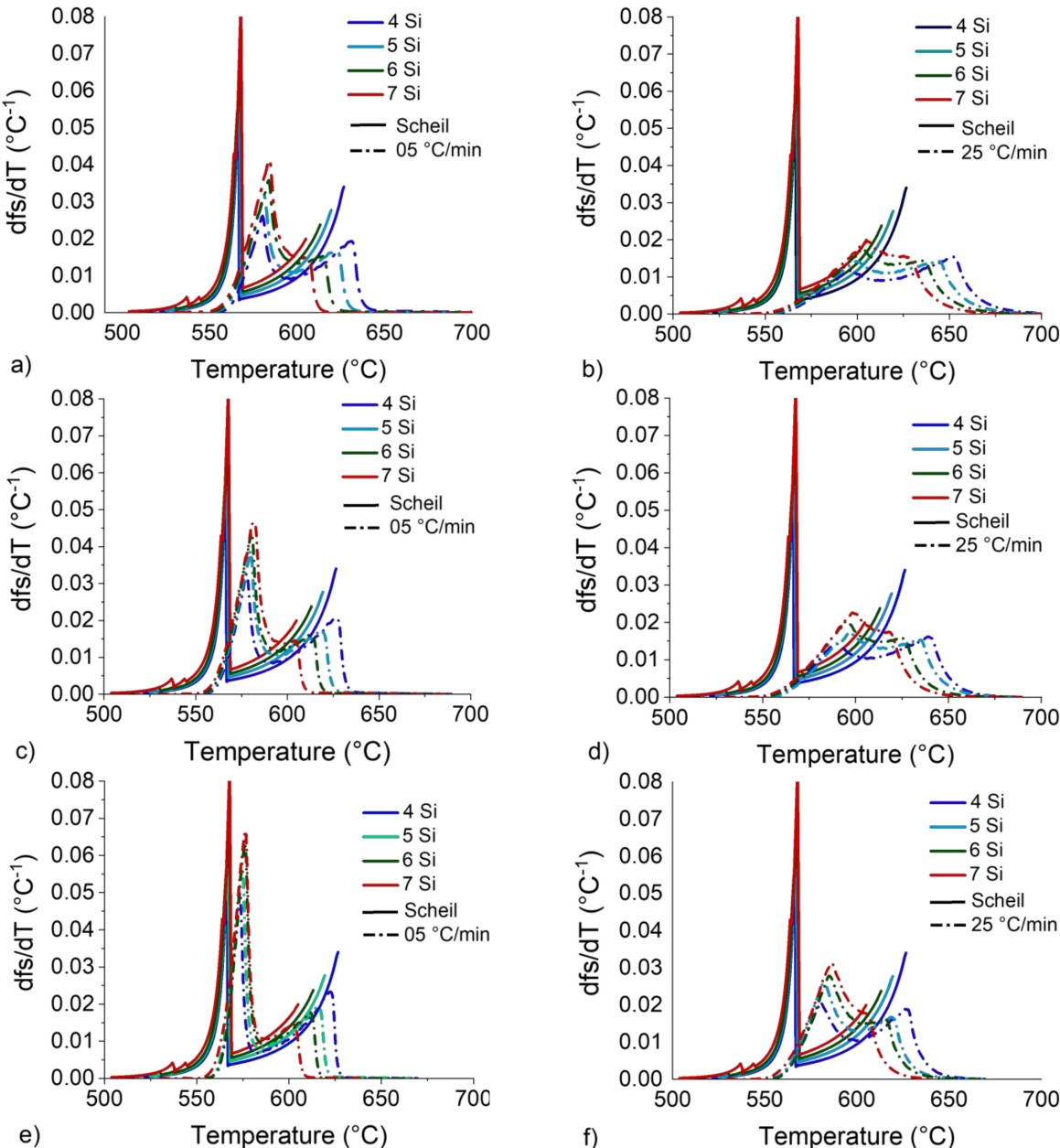

**Figure 14.** $df_S/dT$ vs. $T$ curves for 4 to 7 wt.%Si obtained by CALPHAD simulation at the Scheil condition and DSC curves for heating rates of 5 (left) and 25 °C/min (right) and DSC sample masses of 200 (**a,b**), 90 (**c,d**) and 20 mg (**e,f**).

Since high Si content represented a limitation when identifying $\Delta T_{THIXO}$ by the DM, alloys with a high Si content should be analyzed with lower DSC masses and heating rates to ensure a suitable processing interval. However, common thixoforming operations usually involve high heating rates (in the order of 50 to 100 °C/min) during partial melting treatment prior to forming. Another methodology for determining $\Delta T_{THIXO}$ should therefore be used with high-Si-content alloys at heating rates similar to those used in industry (i.e., higher heating rates) in an attempt to overcome the limitations discussed here.

The thixoforming working window (both temperature and solid fraction) for the lower-mass samples (20 mg) is shown in Table 9. As discussed above, these are considered the most accurate results for the range of masses tested because of the homogeneous heat transfer along the sample, the possibility of identifying critical temperatures by the DM and the similarity between the experimental and CALPHAD data. The alloys containing 4, 5 and 6 wt.%Si have suitable working windows even at the highest heating rate tested, but when the Si content is increased to 7 wt.%, the working window cannot be defined for 25 °C/min, which is a low heating rate compared with those usually employed for thixoforming operations. The design process for this alloy should therefore take into account alternative heating cycles that enable the phase transformations during partial melting to be more accurately identified.

**Table 9.** Thixoforming working window temperatures (in °C) and solid fractions (in mass fraction) for the Al-Si-Zn alloys studied here for 20 mg samples. Values that were not identified are shown as NI.

| $dT/dt$ (°C/min) | $T_{SSML}$ | | $T_{SSMH}$ | | $\Delta T_{THIXO}$ | | $fs_{SSML}$ | | $fs_{SSMH}$ | | $\Delta fs_{THIXO}$ | |
|---|---|---|---|---|---|---|---|---|---|---|---|---|
| | 5 | 25 | 5 | 25 | 5 | 25 | 5 | 25 | 5 | 25 | 5 | 25 |
| 4Si | 582 | 600 | 622 | 627 | 40 | 27 | 0.58 | 0.51 | 0.07 | 0.13 | 0.51 | 0.38 |
| 5Si | 584 | 602 | 616 | 619 | 32 | 17 | 0.45 | 0.37 | 0.04 | 0.12 | 0.41 | 0.25 |
| 6Si | 586 | 605 | 612 | 615 | 26 | 10 | 0.36 | 0.28 | 0.04 | 0.13 | 0.32 | 0.15 |
| 7Si | 588 | NI | 602 | NI | 14 | NI | 0.23 | NI | 0.05 | NI | 0.18 | NI |

## 4. Conclusions

The effects of Si content, heating rate and sample mass on determination of the thixoforming working window ($\Delta T_{THIXO}$) by DSC were analyzed. Lower values of Si content, heating rate and DSC sample mass allowed the lower and upper limits of $\Delta T_{THIXO}$ to be identified more accurately because of the resulting sharper DSC curves, leading to a complete curvature inversion at the Si-eutectic/Al-$\alpha$ phase transition and, consequently, a DSC derivative curve that crossed the zero-baseline used by the DM. Furthermore, lower sample masses and heating rates resulted in data closer to those obtained by the CALPHAD simulation for all the values of Si content analyzed.

Larger masses were associated with significant heterogeneity in heat transfer flow through the DSC sample, leading to the same phase transformation occurring at different places and times in the sample. The sum of the contributions of each "portion" of phase transformation led to results similar to those for a diffuse transition. An increase in heating rate enhanced this effect by promoting a more critical time-dependent heat flow response to the power supplied and led to an even more diffuse-like bulk transformation. Diffuse transformations resulted in softened curves on which critical points could not be satisfactorily identified by the DM.

Since higher Si content represented a limitation when identifying $\Delta T_{THIXO}$ by the DM, alloys with high Si content should be analyzed with lower DSC masses and heating rates, or another methodology for determining $\Delta T_{THIXO}$ should be used to identify a suitable interval at conditions near thixoforming operations.

**Author Contributions:** D.V.T., G.L.B. and J.R.d.O. Methodology, validation, formal analysis, investigation, writing—original draft preparation; F.M. writing—review and editing, visualization; E.J.Z. Conceptualization; methodology; software; validation; formal analysis; resources; data curation; writing—review and editing; visualization; supervision; project administration; funding acquisition. All authors have read and agreed to the published version of the manuscript.

**Funding:** The authors would like to thank the Brazilian research funding agencies FAPESP (São Paulo Research Foundation—projects 2015/22143-3 and 2018/11802-4), CNPq (National Council for Scientific and Technological Development—project PQ303299/2021-5) and CAPES (Federal Agency for the Support and Improvement of Higher Education—notice 23/2016, process 88881.131045/2016-01) for providing financial support for this study.

**Institutional Review Board Statement:** Not applicable.

**Informed Consent Statement:** Not applicable.

**Data Availability Statement:** Data is contained within the article.

**Acknowledgments:** The authors are also indebted to the Faculty of Mechanical Engineering at University of Oriente and the Faculty of Mechanical Engineering at the University of Campinas for the practical support very kindly provided.

**Conflicts of Interest:** The authors declare no conflict of interest.

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
