# Peer review of "The Thixoforming Process Window for Al-Si-Zn Alloys Using the Differentiation Method: The Role of Si, Heating Rate and Sample Mass"

_metals, doi:10.3390/met12050734_

Round 1

Reviewer 1 Report

Before the Editor makes a decision, I suggest that the authors must take into account the following corrections:

  1. I think the title needs to be reformulated to become more “friendly”.
  2. It is not clear how were obtained the data from Tables.
  3. The meaning of \Delta in equations (2) and (3) is not specified.
  4. From where were taken the data used in the graphic representations?
  5. Some editing "glitches" need to be corrected.
  6. Punctuations are used randomly. Insert comma or full stop after each and every equation accordingly.
  7. Many notions and results are "borrowed" from different already published paper. As such, I think the authors need to emphasize more clearly the contribution of the manuscript from a scientific point of view.
  8. References are not uniformly written. In some references the name of the journal is incorrect abbreviated.
  9. Also, I think, the author must strengthen the References section with some articles that use some similar techniques, to make the techniques used more plausible, for instance: An extension of the domain of influence theorem for generalized thermoelasticity of anisotropic material with voids, J. Comput. Theor. Nanosci., 12(8), 1594-1598, 2015; A novel model of plane waves of two-temperature fiber-reinforced thermoelastic medium under the effect of gravity with three-phase-lag model, Int J Numer Method H, 29(12), 4788-4806, 2019.

 If the authors take into account all these corrections, then this manuscript deserves to be published.  

Author Response

Reviewer 1:

R1.1: I think the title needs to be reformulated to become more “friendly”.

New title:

“The thixoforming process window for Al-Si-Zn alloys using the Differentiation Method: The role of Si, heating rate and sample mass.”

Response: Agreed. The title has changed.

R1.2: It is not clear how were obtained the data from the Tables.

Response: All data are obtained from Figure 1, which shows the heat flow rate as a function of temperature for all conditions tested. In this way, the area under each curve is first calculated, Table 2, and with that, the energy of the transformation, Table 3, then the data obtained from the differentiation of each curve, Table 4, is presented, and with these data, the following tables depending on the equations presented. The paper has 9 tables, we will explain below:

Table 1. AlXSi4Zn alloys (composition in wt%) for use in thixoforming processes; standard un-certainties are shown as the deviation corresponding to a 0.95 confidence interval.

The chemical composition was determined with a BILL OES optical spectrophotometer. (Sentence 61, 62, 63).

Table 2. The area between the DSC curves and baseline for the alloys studied here when heated at 5 and 25 °C/min with different sample masses.

The integral of the HFR (in mW/mg) with respect to temperature (in °C) gives the area (in mW°C/mg) between the DSC curves and a baseline (represented here as HFR vs. T). (Sentence 110, 111)

Table 3. Enthalpy change for the melting transformation for the alloys studied here when heated at 5 and 25 °C/min with different sample masses.

The integral of the HFR (in mW/mg) with respect to time (in s) gives the enthalpy change, ΔH (in J/g), corresponding to the melting transformation.

(Sentence 112, 113)

Table 4. Solidus, liquidus and semisolid interval for the melting transformation for the alloys studied here when heated at 5 and 25 °C/min with different sample masses

Solidus, liquidus, and semisolid intervals are determined using a baseline = 0 established by the differentiation method (DM) [10], deriving the HFR curve with respect to T.

Table 5. Comparison of the variables area between DSC curve and baseline (A in W°C/g), enthalpy change (ΔH in V/g), solidus (S in °C), liquidus (L in °C), and semisolid interval (S-L in °C) for extreme (lower and upper) values of Si content, heating rate, and sample mass.

Ratio between two extreme (lower and upper) conditions, e.g., for the area between the DSC curve and baseline the ratio between Si contents is given by A7Si/A4Si; between heating rates, the ratio is given by A25°C/min/A5°C/min; between sample masses, by A200mg/A20mg; and so on for the other parameters

(Sentence 154, 155, 156)

Table 6. Average DSC sample dimensions

Table 6 shows the average dimensions of the samples weighing 20, 90, and 200 mg. The surface area (As) was considered to be the effective area where a significant amount of heat exchange with the environment occurred (i.e., the flat top and the side of the cylindrical samples).

Table 7. The ∆TTHIXO AlXSi4Zn for 05°C/min and 25 °C/min, with a sample mass of 200 mg, 90 mg, and 20 mg

Starting with the original HFR curve (black curve in Fig. 5.a), the derivative of the HFR curve with respect to T is plotted (red curve in Fig. 5.a) and the temperatures of the beginning (TSSML) and end (TSSMH) of ΔTTHIXO are determined using a baseline = 0 established by the differentiation method (DM) [10]. (Sentence 334-337)

Table 8. The TSSML for AlXSi4Zn in the Sheil condition, 05°C/min and 25 °C/min, with a sample mass of 200 mg, 90 mg, and 20 mg

The temperatures of the beginning (TSSML) and end (TSSMH) are determined using a baseline = 0 established by the differentiation method (DM) [10]. The temperature TSSML  in the Sheil condition are determined in the fS vs. T curves obtained by CALPHAD simulation at the near-equilibrium Scheil condition for 4 to 7 wt.%Si

Table 9. Thixoformiing working window temperatures (in °C) and solid fractions (in mass fraction) for the Al-Si-Zn alloys studied here for 20 mg samples. Values that were not identified are shown as NI.

The positions on the fS curve corresponding to TSSML and TSSMH on the dHFR/dT vs. T curve provide the solid fractions corresponding to these temperatures, i.e., fsSSML and fsSSMH (Fig. 5.c)

R1.3: The meaning of \Delta in equations (2) and (3) is not specified.

Response: …where L and S are the solidus and liquidus and ∆TSL is the semisolid interval in Eqs. 2 to 5…(sentence 229)

R1.4: From where taken the data used in the graphic representations?

Response: The data were obtained from the each curve shown in Figure 1, the explanation is in the Materials and Methods - Sentences 69 to 113.

R1.5: Some editing "glitches" need to be corrected.

Response: Thank you, Sentences 115 to 123 are duplicated in 180 - 188.

R1.6: Punctuations are used randomly. Insert a comma or full stop after each and every equation accordingly.

Response: All sentences and equations were corrected.

R1.7: Many notions and results are "borrowed" from the different already published papers. As such, I think the authors need to emphasize more clearly the contribution of the manuscript from a scientific point of view.

Response: Sentences in Introduction were changed.

In the processing of materials in the semi-solid state, the main issue to take into account is the thermodynamic control capacity of the operations.

In recent years, different criteria have been developed to assess whether the alloy of interest is applicable for semi-solid processing, in other words, whether there is an acceptable process window. Important contributions of thixoformability criteria include the works of Liu et al. [1], Kazakov [2], Tzimas and Zavaliangos [3], Zoqui et al. [4], and recently Hu, X.G et al.[5]. These criteria are divided into three types: the temperature sensitivity of the liquid (solid) fraction [2], the time sensitivity of the liquid fraction [6], and the enthalpy sensitivity of the liquid fraction [5]. However, when it comes to specific alloys, neither enthalpy sensitivity nor time sensitivity provides a clear basis for judgment [7]. In this sense, the temperature sensitivity of the liquid (solid) fraction, to date, is the most used [7]

This paper uses the temperature sensitivity criterion modified by Brollo et al. [15,16] called the Differentiation Method (DM), applicable for both thixoforming and rheocasting. The AlXSi4Zn system is studied with the main objective of increasing the range of raw materials for the semi-solid. As reported, the new alloys are analyzed via DSC under specific conditions that modify (Si content, heating rate, sample mass) the semisolid process window. Therefore, the paper seeks to report the main factors that alter the processing window of the new alloys.

R1.8: References are not uniformly written. In some references, the name of the journal is incorrect and abbreviated.

Response: The references were corrected with the help of Endnote, specifically with the MDPI ACS style, indicated in the instructions for the authors

R1.9: Also, I think, the author must strengthen the References section with some articles that use some similar techniques, to make the techniques used more plausible, for instance: An extension of the domain of influence theorem for generalized thermoelasticity of anisotropic material with voids, J. Comput. Theor. Nanosci., 12(8), 1594-1598, 2015; A novel model of plane waves of the two-temperature fiber-reinforced thermoelastic medium under the effect of gravity with the three-phase-lag model, Int J Numer Method H, 29(12), 4788-4806, 2019.

Response: At least four references were added. all describe the main criteria to define the processing window for semi-solid applications

Reviewer 2 Report

  1. In summary, it is necessary to indicate the scientific purpose of the article and the premises for the analysis of the issue. Moreover, the scientific methods used in the work should be mentioned and the achievement should be indicated. The abstract should be completely changed, and the conclusions should be brought to the last chapter.
  2. The paper presents the influence of a number of factors on the melting range of aluminium alloys. There is no broader analysis of the sensitivity of this range to the change of the analyzed parameters. It should be absolutely added.
  3. The authors should indicate in this work the criteria of thixoformability of aluminium alloys (according the technological possibilities) and determine the range of analyzed factors.
  4. The title of the article should be changed and “the melting range” should be used instead of “the thixoforming working window”. Not all aluminium alloys can be formed in the semi-solid state.
  5. Article written carelessly. Some Figures in the text have no numbers.
  6. In the thermodynamic calculations using the Calphad sotware the kinetics models should be used instead of the Scheil model. They are more accurate. The solid fraction should be recalculated.

Author Response

Reviewer 2

R2.1: In summary, it is necessary to indicate the scientific purpose of the article and the premises for the analysis of the issue. Moreover, the scientific methods used in the work should be mentioned and the achievement should be indicated. The abstract should be completely changed, and the conclusions should be brought to the last chapter.

Response: The sentence "In the processing of materials in the semi-solid state, the main issue to take into account is the thermodynamic control capacity of the operations".

R2.2: The paper presents the influence of a number of factors on the melting range of aluminium alloys. There is no broader analysis of the sensitivity of this range to the change of the analyzed parameters. It should be absolutely added.

Response: The influence of each factor is presented in items 3.1 to 3.5.

R2.3: The authors should indicate in this work the criteria of thixoformability of aluminium alloys (according the technological possibilities) and determine the range of analyzed factors.

Response: the thixoformability criteria was added, att the sentence  In this sense, the temperature sensitivity of the liquid (solid) fraction (dfL/dT or dfS/dT in oC-1), to date, is the most used [7]. Therefore the criteria adopted in this paper is as follow: "The sensitivity of the liquid fraction (or solid fraction - dfL/dT or dfL/dT), at the desired liquid fraction, (or solid fraction - fL or fS), exclusively for the primary phase must be as low as possible (<0.03°C-1)" [4].

R2.4: The title of the article should be changed and “the melting range” should be used instead of “the thixoforming working window”. Not all aluminium alloys can be formed in the semi-solid state.

Response: The title was changed.

R2.5: Article written carelessly. Some Figures in the text have no numbers.

Response: The text and figure were corrected.

R2.6: In the thermodynamic calculations using the Calphad software the kinetics models should be used instead of the Scheil model. They are more accurate. The solid fraction should be recalculated.

Response: Unfortunately the authors have only a limited edition of the ThermoCalc Software and only the solution via Scheil equation is available.

The authors hope to have corrected all the flaws raised by the reviewers

                    Eugênio José Zoqui

Round 2

Reviewer 1 Report

The manuscript has been improved, but not completely. The authors must answer all my questions.

Author Response

In order to better clarify, the file with the answers is attached
